# Information-theoretic lower bounds for convex optimization with erroneous oracles

**Yaron Singer**
Harvard University
Cambridge, MA 02138
yaron@seas.harvard.edu

**Jan Vondrák**
IBM Almaden Research Center
San Jose, CA 95120
jvondrak@us.ibm.com

## Abstract

We consider the problem of optimizing convex and concave functions with access to an erroneous zeroth-order oracle. In particular, for a given function $\mathbf{x} \to f(\mathbf{x})$ we consider optimization when one is given access to *absolute error* oracles that return values in $[f(\mathbf{x}) - \epsilon, f(\mathbf{x}) + \epsilon]$ or *relative error* oracles that return value in $[(1 - \epsilon)f(\mathbf{x}), (1 + \epsilon)f(\mathbf{x})]$, for some $\epsilon > 0$. We show stark information theoretic impossibility results for minimizing convex functions and maximizing concave functions over polytopes in this model.

## 1 Introduction

Consider the problem of minimizing a convex function over some convex domain. It is well known that this problem is solvable in the sense that there are algorithms which make polynomially-many calls to an oracle that evaluates the function at every given point, and return a point which is arbitrarily close to the true minimum of the function. But suppose that instead of the true value of the function, the oracle has some small error. Would it still be possible to optimize the function efficiently? To formalize the notion of error, we can consider two types of erroneous oracles:

- For a given function $f : [0,1]^n \to [0,1]$ we say that $\widetilde{f} : [0,1]^n \to [0,1]$ is an **absolute** $\epsilon$-erroneous oracle if $\forall \mathbf{x} \in [0,1]^n$ we have that: $\widetilde{f}(\mathbf{x}) = f(\mathbf{x}) + \xi_{\mathbf{x}}$ where $\xi_{\mathbf{x}} \in [-\epsilon, \epsilon]$.

- For a given function $f : [0,1]^n \to \mathbb{R}$ we say that $\widetilde{f} : [0,1]^n \to \mathbb{R}$ is a **relative** $\epsilon$-erroneous oracle if $\forall \mathbf{x} \in [0,1]^n$ we have that: $\widetilde{f}(\mathbf{x}) = \xi_{\mathbf{x}} f(x)$ where $\xi_{\mathbf{x}} \in [1 - \epsilon, 1 + \epsilon]$.

Note that we intentionally do not make distributional assumptions about the errors. This is in contrast to *noise*, where the errors are assumed to be random and independently generated from some distribution. In such cases, under reasonable conditions on the distribution, one can obtain arbitrarily good approximations of the true function value by averaging polynomially many points in some $\epsilon$-ball around the point of interest. Stated in terms of *noise*, in this paper we consider oracles that have some small *adversarial noise*, and wish to understand whether desirable optimization guarantees are obtainable. To avoid ambiguity, we refrain from using the term *noise* altogether, and refer to such as inaccuracies in evaluation as *error*.

While distributional i.i.d. assumptions are often reasonable models, evaluating our dependency on these assumptions seems necessary. From a practical perspective, there are cases in which noise can be correlated, or where the data we use to estimate the function is corrupted in some arbitrary way. Furthermore, since we often optimize over functions that we learn from data, the process of fitting a model to a function may also introduce some bias that does not necessarily vanish, or may vanish. But more generally, it seems like we should morally know the consequences that modest inaccuracies may have.

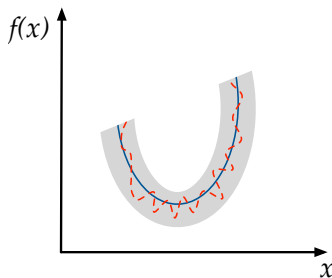

Figure 1: An illustration of an erroneous oracle to a convex function that fools a gradient descent algorithm.

**Benign cases.** In the special case of a linear function $f(\mathbf{x}) = \mathbf{c}^\intercal \mathbf{x}$, for some $\mathbf{c} \in \mathbb{R}^n$, a relative $\epsilon$-error has little effect on the optimization. By querying $f(\mathbf{e}_i)$, for every $i \in [n]$ we can extract $\widetilde{c}_i \in [(1-\epsilon)c_i, (1+\epsilon)c_i]$ and then optimize over $f'(\mathbf{x}) = \widetilde{\mathbf{c}}^\intercal \mathbf{x}$. This results in a $(1\pm\epsilon)$-multiplicative approximation. Alternatively, if the erroneous oracle $\widetilde{f}$ happens to be a convex function, optimizing $\tilde{f}(\mathbf{x})$ directly retains desirable optimization guarantees, up to either additive and multiplicative errors. We are therefore interested in scenarios where the error does not necessarily have nice properties.

**Gradient descent fails with error.** For a simple example, consider the function illustrated in Figure 1. The figure illustrates a convex function (depicted in blue) and an erroneous version of it (dotted red), s.t. on every point, the oracle is at most some additive $\epsilon > 0$ away from the true function value (the $\epsilon$ margins of the function are depicted in grey). If we assume that a gradient descent algorithm is given access to the erroneous version (dotted red) instead of the true function (blue), the algorithm will be trapped in a local minimum that can be arbitrarily far from the true minimum. But the fact that a naive gradient descent algorithm fails does not necessarily mean that there isn't an algorithm that can overcome small errors. This narrates the main question in this paper.

*Is convex optimization robust to error?*

**Main Results.** Our results are largely spoilers. We present stark information-theoretic lower bounds for both relative and absolute $\epsilon$-erroneous oracles, for any constant and even sub-constant $\epsilon > 0$. In particular, we show that:

- For minimizing a convex function (or maximizing a concave function) $f : [0,1]^n \to [0,1]$ over $[0,1]^n$: we show that for any fixed $\delta > 0$, no algorithm can achieve an additive approximation within $1/2 - \delta$ of the optimum, using a subexponential number of calls to an absolute $n^{-1/2+\delta}$-erroneous oracle.

- For minimizing a convex function $f : [0,1]^n \to [0,1]$ over a polytope $\mathcal{P} \subset [0,1]^n$: for any fixed $\epsilon > 0$, no algorithm can achieve a finite multiplicative factor using a subexponential number of calls to a relative $\epsilon$-erroneous oracle.

- For maximizing a concave function $f : [0,1]^n \to [0,1]$ over a polytope $\mathcal{P} \subset [0,1]^n$: for any fixed $\epsilon > 0$, no algorithm can achieve a multiplicative factor better than $\Theta(n^{-1/2+\epsilon})$ using a subexponential number of calls to a relative $\epsilon$-erroneous oracle;

- For maximizing a concave function $f : [0,1]^n \to [0,1]$ over $[0,1]^n$: for any fixed $\epsilon > 0$, no algorithm can obtain a multiplicative factor better than $1/2 + \epsilon$ using a subexponential number of calls to a relative $\epsilon$-erroneous oracle. (And there is a trivial $1/2$-approximation without asking any queries.)

Somewhat surprisingly, many of the impossibility results listed above are shown for a class of extremely simple convex and concave functions, namely, affine functions: $f(\mathbf{x}) = \mathbf{c}^\intercal \mathbf{x} + b$. This is

in sharp contrast to the case of linear functions (without the constant term $b$) with relative erroneous oracles as discussed above. In addition, we note that our results extend to strongly convex functions.

## 1.1 Related work

The oracle models we study here fall in the category of *zeroth-order* or *derivative free*. Derivative-free methods have a rich history in convex optimization and were among the earliest to numerically solve unconstrained optimization problems. Recently these approaches have enjoyed increasing interest, as they are useful in scenarios where black-box access is given to the function or cases in which gradient information is difficult to compute or does not exist [9, 8, 11, 15, 14, 6]

There has been a rich line of work for noisy oracles, where the oracles return some erroneous version of the function value which is random. In a stochastic framework, these settings correspond to repeatedly choosing points in some convex domain, obtaining noisy realizations of some underlying convex function's value. Most frequently, the assumption is that one is given a first-order noisy oracle with some assumptions about the distribution that generates the error [13, 12]. In the learning theory community, optimization with stochastic noisy oracles is often motivated by multi-armed bandits settings [4, 1], and regret minimization with zeroth-order feedback [2]. All these models consider the case in which the error is drawn from a distribution.

The model of adversarial noise in zeroth order oracles has been mentioned in [10] which considers a related model of erroneous oracles and informally argues that exponentially many queries are required to approximately minimize a convex function in this model (under an $\ell_2$-ball constraint).

In recent work, Belloni et al. [3] study convex optimization with erroneous oracles. Interestingly, Belloni et al. show positive results. In their work they develop a novel algorithm that is based on sampling from an approximately log-concave distribution using the Hit-and-Run method and show that their method has polynomial query complexity. In contrast to the negative results we show in this work, the work of Belloni et al. assumes the (absolute) erroneous oracle returns $f(\mathbf{x}) + \xi_{\mathbf{x}}$ with $\xi_{\mathbf{x}} \in [-\frac{\epsilon}{n}, \frac{\epsilon}{n}]$. That is, the error is not a constant term, but rather is inversely proportional to the dimension. Our lower bounds for additive approximation hold when the oracle error is not necessarily a constant but $\xi_{\mathbf{x}} \in [\frac{1}{n^{1/2-\delta}}, \frac{1}{n^{1/2-\delta}}]$ for a constant $0 < \delta < 1/2$.

## 2 Preliminaries

**Optimization and convexity.** For a minimization problem, given a nonnegative objective function $f$ and a polytope $\mathcal{P}$ we will say that an algorithm provides a (multiplicative) $\alpha$-approximation ($\alpha > 1$) if it finds a point $\bar{\mathbf{x}} \in \mathcal{P}$ s.t. $f(\bar{\mathbf{x}}) \leq \alpha \min_{\mathbf{x} \in \mathcal{P}} f(\mathbf{x})$. For a maximization problem, an algorithm provides an $\alpha$-approximation ($\alpha < 1$) if it finds a point $\bar{\mathbf{x}}$ s.t. $f(\bar{\mathbf{x}}) \geq \alpha \max_{\mathbf{x} \in \mathcal{P}} f(\mathbf{x})$.

For absolute erroneous oracles, given an objective function $f$ and a polytope $\mathcal{P}$ we will aim to find a point $\bar{\mathbf{x}} \in \mathcal{P}$ which is within an additive error of $\delta$ from the optimum, with $\delta$ as small as possible. That is, for a $\delta > 0$ we aim to find a point $\bar{\mathbf{x}}$ s.t. $|f(\bar{\mathbf{x}}) - \min_{\mathbf{x}} f(\mathbf{x})| < \delta$ in the case of minimization.

A function $f : \mathcal{P} \to \mathbb{R}$ is convex on $\mathcal{P}$ if $f(t\mathbf{x} + (1-t)\mathbf{y}) \leq tf(\mathbf{x}) + (1-t)f(\mathbf{y})$ (or concave if $f(t\mathbf{x} + (1-t)\mathbf{y}) \geq tf(\mathbf{x}) + (1-t)f(\mathbf{y})$) for every $\mathbf{x}, \mathbf{y} \in \mathcal{P}$ and $t \in [0, 1]$.

**Chernoff bounds.** Throughout the paper we appeal to the Chernoff bounds. We note that while typically stated for independent random variables $X_1, \ldots, X_m$, Chernoff bounds also hold for *negatively associated* random variables.

**Definition 2.1** ([5], Definition 1). *Random variables $X_1, \ldots, X_n$ are negatively associated, if for every $I \subseteq [n]$ and every non-decreasing $f : \mathbb{R}^I \to \mathbb{R}, g : \mathbb{R}^{\bar{I}} \to \mathbb{R}$,*

$$\mathsf{E}[f(X_i, i \in I)g(X_j, j \in \bar{I})] \leq \mathsf{E}[f(X_i, i \in I)]\mathsf{E}[g(X_j, j \in \bar{I})].$$

**Claim 2.2** ([5], Theorem 14). *Let $X_1, \ldots, X_n$ be negatively associated random variables that take values in $[0, 1]$ and $\mu = \mathbb{E}[\sum_{i=1}^{n} X_i]$. Then, for any $\delta \in [0, 1]$ we have that:*

$$\Pr[\sum_{i=1}^{n} X_i > (1 + \delta)\mu] \leq e^{-\delta^2 \mu/3},$$

$$\Pr[\sum_{i=1}^{n} X_i < (1 - \delta)\mu] \leq e^{-\delta^2\mu/2}.$$

We apply this to random variables that are formed by selecting a random subset of a fixed size. In particular, we use the following.

**Claim 2.3.** *Let $x_1, \ldots, x_n \geq 0$ be fixed. For $1 \leq k \leq n$, let $R$ be a uniformly random subset of $k$ elements out of $[n]$. Let $X_i = x_i$ if $i \in R$ and $X_i = 0$ otherwise. Then $X_1, \ldots, X_n$ are negatively associated.*

*Proof.* For $x_1 = x_2 = \ldots = x_n = 1$, the statement holds by Corollary 11 of [5] (which refers to this distribution as the Fermi-Dirac model). The generalization to arbitrary $x_i \geq 0$ follows from Proposition 4 of [5] with $I_j = \{j\}$ and $h_j(t) = x_j t$. $\qquad\square$

# 3 Optimization over the unit cube

We start with optimization over $[0, 1]^n$, arguably the simplest possible polytope. We show that already in this setting, the presence of adversarial noise prevents us from achieving much more than trivial results.

## 3.1 Convex minimization

First let us consider convex minimization over $[0, 1]^n$. In this setting, we show that errors as small as $n^{-(1-\delta)/2}$ prevent us from optimizing within a constant additive error.

**Theorem 3.1.** *Let $\delta > 0$ be a constant. There are instances of a convex function $f : [0, 1]^n \to [0, 1]$ accessible through an absolute $n^{-(1-\delta)/2}$-erroneous oracle, such that a (possibly randomized) algorithm that makes $e^{O(n^\delta)}$ queries cannot find a solution of value better than within additive $1/2 - o(1)$ of the optimum with probability more than $e^{-\Omega(n^\delta)}$.*

We remark that the proof of this theorem is inspired by the proof of hardness of $(\frac{1}{2} + \epsilon)$-approximation for unconstrained submodular maximization [7]; in particular it can be viewed as a simple application of the "symmetry gap" argument (see [16] for a more general exposition).

*Proof.* Let $\epsilon = n^{-(1-\delta)/2}$; we can assume that $\epsilon < \frac{1}{2}$, otherwise $n$ is constant and the statement is trivial. We will construct an $\epsilon$-erroneous oracle (both in the relative and absolute sense) for a convex function $f : [0, 1]^n \to [0, 1]$. Consider a partition of $[n]$ into two subsets $A, B$ of size $|A| = |B| = n/2$ (which will be eventually chosen randomly). We define the following function:

- $f(\mathbf{x}) = \frac{1}{2} + \frac{1}{n}(\sum_{i \in A} x_i - \sum_{j \in B} x_j)$.

This is a convex (in fact linear) function. Next, we define the following modification of $f$, which could be the function returned by an $\epsilon$-erroneous oracle.

- If $|\sum_{i \in A} x_i - \sum_{j \in B} x_j| > \frac{1}{2}\epsilon n$, then $\tilde{f}(\mathbf{x}) = f(\mathbf{x}) = \frac{1}{2} + \frac{1}{n}(\sum_{i \in A} x_i - \sum_{j \in B} x_j)$.

- If $|\sum_{i \in A} x_i - \sum_{j \in B} x_j| \leq \frac{1}{2}\epsilon n$, then $\tilde{f}(\mathbf{x}) = \frac{1}{2}$.

Note that $f(\mathbf{x})$ and $\tilde{f}(\mathbf{x})$ differ only in the region where $|\sum_{i \in A} x_i - \sum_{j \in B} x_j| \leq \frac{1}{2}\epsilon n$. In particular, the value of $f(\mathbf{x})$ in this region is within $[\frac{1-\epsilon}{2}, \frac{1+\epsilon}{2}]$, while $\tilde{f}(\mathbf{x}) = \frac{1}{2}$, so an $\epsilon$-erroneous oracle for $f(\mathbf{x})$ (both in the relative and absolute sense) could very well return $\tilde{f}(\mathbf{x})$ instead.

Now assume that $(A, B)$ is a random partition, unknown to the algorithm. We argue that with high probability, a fixed query $\mathbf{x}$ issued by the algorithm will have the property that $|\sum_{i \in A} x_i - \sum_{j \in B} x_j| \leq \frac{1}{2}\epsilon n$. More precisely, since $(A, B)$ is chosen at random subject to $|A| = |B| = n/2$,

we have that $\sum_{i \in A} x_i$ is a sum of negatively associated random variables in $[0,1]$ (by Claim 2.3). The expectation of this quantity is $\mu = \mathsf{E}[\sum_{i \in A} x_i] = \frac{1}{2}\sum_{i=1}^{n} x_i \leq \frac{1}{2}n$. By Claim 2.2, we have

$$\Pr[\sum_{i \in A} x_i > \mu + \frac{1}{4}\epsilon n] = \Pr[\sum_{i \in A} x_i > (1 + \frac{n}{4\mu}\epsilon)\mu] < e^{-(n\epsilon/(4\mu))^2 \mu/3} \leq e^{-\epsilon^2 n/24}.$$

Since $\frac{1}{2}\sum_{i \in A} x_i + \frac{1}{2}\sum_{i \in B} x_i = \frac{1}{2}\sum_{i=1}^{n} x_i = \mu$, we get

$$\Pr[\sum_{i \in A} x_i - \sum_{i \in B} x_i > \frac{1}{2}\epsilon n] = \Pr[\sum_{i \in A} x_i - \mu > \frac{1}{4}\epsilon n] < e^{-\epsilon^2 n/24}.$$

By symmetry,

$$\Pr[|\sum_{i \in B} x_i - \sum_{j \in A} x_j| > \frac{1}{2}\epsilon n] < 2e^{-\epsilon^2 n/24}.$$

We emphasize that this holds for a *fixed query* $\mathbf{x}$.

Recall that we assumed the algorithm to be deterministic. Hence, as long as its queries satisfy the property above, the answers will be $\tilde{f}(\mathbf{x}) = 1/2$, and the algorithm will follow the same path of computation, no matter what the choice of $(A, B)$ is. (Effectively we will not learn anything about $A$ and $B$.) Considering the sequence of queries on this computation path, if the number of queries is $t$ then with probability at least $1 - 2te^{-\epsilon^2 n/24}$ the queries will indeed fall in the region where $\tilde{f}(\mathbf{x}) = 1/2$ and the algorithm will follow this path. If $t \leq e^{\epsilon^2 n/48}$, this happens with probability at least $1 - 2e^{-\epsilon^2 n/48}$. In this case, all the points queried by the algorithm as well as the returned solution $\mathbf{x}_{out}$ (by the same argument) satisfies $\tilde{f}(\mathbf{x}_{out}) = 1/2$, and hence $f(\mathbf{x}_{out}) \geq \frac{1-\epsilon}{2}$. In contrast, the actual optimum is $f(\mathbb{1}_B) = 0$. Recall that $\epsilon = n^{-(1-\delta)/2}$; hence, $f(\mathbf{x}_{out}) \geq \frac{1}{2}(1 - n^{-(1-\delta)/2})$ and the bounds on the number of queries and probability of success are as in the statement of the theorem.

Finally, consider a randomized algorithm. Denote by $(R_1, R_2, \ldots, \ldots)$ the random variables used by the algorithm in its decisions. We can condition on a fixed choice of $(R_1 = r_1, R_2 = r_2, \ldots)$ which makes the algorithm deterministic. By our proof, the algorithm conditioned on this choice cannot succeed with probability more than $e^{-\Omega(n^\delta)}$. Since this is true for each particular choice of $(r_1, r_2, \ldots)$, by averaging it is also true for a random choice of $(R_1, R_2, \ldots)$. Hence, we obtain the same result for randomized algorithms as well. $\qquad\square$

## 3.2 Concave maximization

Here we consider the problem of maximizing a concave function $f : [0,1]^n \to [0,1]$. One can obtain a result for concave maximization analogous to Theorem 3.1, which we do not state; in terms of additive errors, there is really no difference between convex minimization and concave maximization. However, in the case of concave maximization we can also formulate the following hardness result for multiplicative approximation.

**Theorem 3.2.** *If a concave function $f : [0,1]^n \to [0,1]$ is accessible through a relative-$\delta$-erroneous oracle, then for any $\epsilon \in [0, \delta]$, an algorithm that makes less than $e^{\epsilon^2 n/48}$ queries cannot find a solution of value greater than $\frac{1+\epsilon}{2} OPT$ with probability more than $2e^{-\epsilon^2 n/48}$.*

*Proof.* This result follows from the same construction as Theorem 3.1. Recall that $f(\mathbf{x})$ is a linear function, hence also concave. As we mentioned in the proof of Theorem 3.1, $\tilde{f}(\mathbf{x})$ could be the values returned by a relative $\epsilon$-erroneous oracle. Now we consider an arbitrary $\epsilon > 0$; note that for $\delta \geq \epsilon$ it still holds that $\tilde{f}(\mathbf{x})$ is a relative $\delta$-erroneous oracle.

By the same proof, an algorithm querying less than $e^{\epsilon^n n/48}$ points cannot find a solution of value better than $\frac{1+\epsilon}{2}$ with probability more than $2e^{-\epsilon^2 n/48}$. In contrast, the optimum of the maximization problem is $\sup_{\mathbf{x} \in [0,1]^n} f(\mathbf{x}) = 1$. Therefore, the algorithm cannot achieve multiplicative approximation better than $\frac{1+\epsilon}{2}$. $\qquad\square$

We note that this hardness result is optimal due to the following easy observation.

**Theorem 3.3.** *For any concave function $f : [0,1]^n \to \mathbb{R}_+$, let $OPT = \sup_{\mathbf{x} \in [0,1]^n} f(\mathbf{x})$. Then*

$$f\left(\frac{1}{2}, \frac{1}{2}, \dots, \frac{1}{2}\right) \geq \frac{1}{2} OPT.$$

*Proof.* By compactness, the optimum is attained at a point: let $OPT = f(\mathbf{x}^*)$. Let also $\mathbf{x}' = (1, 1, \dots, 1) - \mathbf{x}^*$. We have $\mathbf{x}' \in [0,1]^n$ and hence $f(\mathbf{x}') \geq 0$. By concavity, we obtain

$$f\left(\frac{1}{2}, \frac{1}{2}, \dots, \frac{1}{2}\right) = f\left(\frac{\mathbf{x}^* + \mathbf{x}'}{2}\right) \geq \frac{f(\mathbf{x}^*) + f(\mathbf{x}')}{2} \geq \frac{1}{2} f(\mathbf{x}^*) = \frac{1}{2} OPT.$$

$\square$

In other words, a multiplicative $\frac{1}{2}$-approximation for this problem is trivial to obtain — even without asking any queries about $f$. We just return the point $(\frac{1}{2}, \frac{1}{2}, \dots, \frac{1}{2})$. Thus we can conclude that for concave maximization, a relative $\epsilon$-erroneous oracle is not useful at all.

## 4 Optimization over polytopes

In this section we consider optimization of convex and concave functions over a polytope $\mathcal{P} = \{\mathbf{x} \geq 0 \ : \ A\mathbf{x} = b\}$. We will show inappoximability results for the relative error model. Note that for the absolute error case, the lower bound on convex minimization from the previous section holds, and can be applied to show a lower bound for concave maximization with absolute errors.

**Theorem 4.1.** *Let $\epsilon, \delta \in (0, 1/2)$ be some constants. There are convex functions for which no algorithm can obtain a finite approximation ratio to $\min_{\mathbf{x} \in \mathcal{P}} f(\mathbf{x})$ using $\Omega(e^{n^\delta})$ queries to a relative $\epsilon$-erroneous oracle of the function.*

*Proof.* We will prove our theorem for the case in which $\mathcal{P} = \{\mathbf{x} \geq 0 \ : \ \sum_i x_i \leq n^{1/2+\delta}\}$. Let $H$ be a subset of indices chosen uniformly at random from all subsets of size exactly $n^{1/2+\delta}$. We construct two functions:

$$f(\mathbf{x}) = n^{1+\delta} - n^{1/2} \sum_{i \in H} x_i$$

$$g(\mathbf{x}) = n^{1+\delta} - n^\delta \sum_i x_i$$

Observe that both these functions are convex and non-negative. Also, observe that the minimizer of $f$ is $\mathbf{x}^* = \mathbb{1}_H$ and $f(\mathbf{x}^*) = 0$, while the minimizer of $g$ is any vector $\mathbf{x}' : \sum_i x_i = n^{1/2+\delta}$ and $g(\mathbf{x}') = n^{1+\delta} - n^{1/2+2\delta} = \Theta(n^{1+\delta})$. Therefore, the ratio between these two functions is unbounded. We will now construct the erroneous oracle in the following manner:

$$\widetilde{f}(\mathbf{x}) = \begin{cases} g(\mathbf{x}), & \text{if } (1 - \epsilon)f(\mathbf{x}) \leq g(\mathbf{x}) \leq (1 + \epsilon)f(\mathbf{x}) \\ f(\mathbf{x}) & \text{otherwise} \end{cases}$$

By definition, $\widetilde{f}$ is an $\epsilon$-erroneous oracle to $f$. The claim will follow from the fact that given access to $\widetilde{f}$ one cannot distinguish between $f$ and $g$ using a subexponential number of queries. This implies the inapproximability result since an approximation algorithm which guarantees a finite approximation ratio using a subexponential number of queries could be used to distinguish between the two functions: if the algorithm returns an answer strictly greater than $0$ then we know the underlying function is $g$ and otherwise it is $f$.

Given a query $\mathbf{x} \in [0,1]^n$ to the oracle, we will consider two cases.

- In case the query $\mathbf{x}$ is such that $\sum_i x_i \leq n^{1/2}$ then we have that:

$$n^{1+\delta} - n \leq f(\mathbf{x}) \leq n^{1+\delta}$$

$$n^{1+\delta} - n^{\delta+1/2} \leq g(\mathbf{x}) \leq n^{1+\delta}$$

Since for any $\epsilon, \delta > 0$ there is a large enough $n$ s.t. $n^\delta > (1 + \epsilon)/\epsilon$, this implies that for any query for which $\sum_i x_i \le n^{1/2}$ then we have that $g(\mathbf{x}) \in [(1 - \epsilon)f(\mathbf{x}), (1 + \epsilon)f(\mathbf{x})]$ and thus the oracle returns $g(\mathbf{x})$.

- In case the query is such that $\sum_i x_i > n^{1/2}$ then we can interpret the value of $\sum_{i \in H} x_i$ which determines value of $f$ as a sum of negatively associated random variables $X_1, \ldots, X_n$ where $X_i$ realizes with probability $n^{-1/2+\delta}$ and takes value $x_i$ if realized (see Claim 2.3). We can then apply the Chernoff bound (Claim 2.2), using the fact that $\mathbb{E}[f(\mathbf{x})] = n^{1/2-\delta} \sum_i x_i$, and get that for any constant $0 < \beta < 1$ we have that with probability $1 - e^{-\Omega(n^\delta)}$:

$$\left(1 - \beta\right) \frac{\sum_i x_i}{n^{1/2-\delta}} \le \sum_{i \in H} x_i \le \left(1 + \beta\right) \frac{\sum_i x_i}{n^{1/2-\delta}}$$

By using $\beta \le \epsilon/(1 + \epsilon)$, this implies that with probability at least $1 - e^{-\Omega(n^\delta)}$ we get that:
$$(1 - \epsilon)f(\mathbf{x}) \le g(\mathbf{x}) \le (1 + \epsilon)f(\mathbf{x})$$

Since the likelihood of distinguishing between $f$ and $g$ on a single query is exponentially small in $n^\delta$, the same arguments used throughout the paper imply that it takes an exponential number of queries to distinguish between $f$ and $g$.

To conclude, for any query $\mathbf{x} \in [0, 1]^n$ it takes $\Omega(e^{n^\delta})$ queries to distinguish between $f$ and $g$. As discussed above, due to the fact that the ratio between the optima of these two functions is unbounded, this concludes the proof. □

**Theorem 4.2.** $\forall$ *constants* $\epsilon, \delta \in (0, 1/2)$ *there is a concave function* $f : [0, 1]^n \to \mathbb{R}_+$ *for which no algorithm can obtain an approximation strictly better than* $O(n^{-1/2+\delta})$ *to* $\max_{\mathbf{x} \in \mathcal{P}} f(\mathbf{x})$ *using* $\Omega(e^{n^\delta})$ *queries to a relative* $\epsilon$-*erroneous oracle of the function.*

*Proof.* We follow a similar methodology as in the proof of Theorem 4.1. We again we select a set $H$ of size $n^{1/2+\delta}$ u.a.r. and construct two functions: $f(\mathbf{x}) = n^{1/2} \sum_{i \in H} x_i + \frac{n^{1/2+\delta}}{\epsilon}$ and $g(\mathbf{x}) = n^\delta \sum_i x_i + \frac{n^{1/2+\delta}}{\epsilon}$. As in the proof of Theorem 4.1 the noisy oracle $\tilde{f}(\mathbf{x}) = g(\mathbf{x})$ when $(1-\epsilon)f(\mathbf{x}) \le g(\mathbf{x}) \le (1+\epsilon)f(\mathbf{x})$ and otherwise $\tilde{f}(\mathbf{x}) = f(\mathbf{x})$. Note that both functions are concave and non-negative, and by its definition the oracle is $\epsilon$-erroneous for the function $f$. For $b = n^{1/2+\delta}$ it is easy to see that the optimal value when the objective is $f$ is $n^{1+\delta}$ while the optimal value is $O(n^{1/2+\delta})$ when the objective is $g$, which implies that one cannot obtain an approximation better than $\Omega(n^{-1/2+\delta})$ with a subexponential number of queries. In case the query to the oracle is a point $\mathbf{x}$ s.t. $\sum_i x_i \le n^{1/2}$, then by Chernoff bound arguments, similar to the ones we used above, with probability at least $1 - e^{-\Omega(n^\delta)}$ we get $(1 - \epsilon)f(\mathbf{x}) \le g(\mathbf{x}) \le (1 + \epsilon)f(\mathbf{x})$. Thus, for any query in which $\sum_i x_i \le n^{1/2}$, the likelihood of the oracle returning $f$ is exponentially small in $n^\delta$.

In case the query is a point $\mathbf{x}$ s.t. $\sum_i x_i > n^{1/2}$ standard concentration bound arguments as before, imply that with probability at least $1 - e^{-\Omega(n^\delta)}$ we get $(1 - \epsilon)f(\mathbf{x}) \le g(\mathbf{x}) \le (1 + \epsilon)f(\mathbf{x})$. Since the likelihood of distinguishing between $f$ and $g$ on a single query is exponentially small in $n^\delta$, we can conclude that it takes an exponential number of queries to distinguish between $f$ and $g$. □

## 5 Optimization over assignments

In this section, we consider the concave maximization problem over a more specific polytope,

$$P_{n,k} = \left\{ \mathbf{x} \in \mathbb{R}_+^{n \times k} : \sum_{j=1}^{k} x_{ij} = 1 \; \forall i \in [n] \right\}.$$

This can be viewed as the matroid polytope for a partition matroid on $n$ blocks of $k$ elements, or alternatively the convex hull of assignments of $n$ items to $k$ agents. In this case, there is a trivial $\frac{1}{k}$-approximation, similar to the $\frac{1}{2}$-approximation in the case of a unit cube.

**Theorem 5.1.** *For any $k \geq 2$ and a concave function $f : P_{n,k} \to \mathbb{R}_+$, let $OPT = \sup_{\mathbf{x} \in P_{n,k}} f(\mathbf{x})$. Then*

$$f\left(\frac{1}{k}, \frac{1}{k}, \ldots, \frac{1}{k}\right) \geq \frac{1}{k}OPT.$$

*Proof.* By compactness, the optimum is attained at a point: let $OPT = f(\mathbf{x}^*)$. Let $x_{ij}^{(\ell)} = x_{i,(j+\ell \bmod k)}^*$ ; i.e., $\mathbf{x}^{(\ell)}$ is a cyclic shift of the coordinates of $\mathbf{x}^*$ by $\ell$ in each block. We have $\mathbf{x}^{(\ell)} \in P_{n,k}$ and $\frac{1}{k}\sum_{\ell=0}^{k-1} x_{ij}^{(\ell)} = \frac{1}{k}\sum_{j=1}^{k} x_{ij}^* = \frac{1}{k}$. By concavity and nonnegativity of $f$, we obtain

$$f\left(\frac{1}{k}, \frac{1}{k}, \ldots, \frac{1}{k}\right) = f\left(\frac{1}{k}\sum_{\ell=0}^{k-1} \mathbf{x}^{(\ell)}\right) \geq \frac{1}{k}f(\mathbf{x}^{(0)}) = \frac{1}{k}OPT.$$

$\square$

We show that this approximation is best possible, if we have access only to a $\delta$-erroneous oracle.

**Theorem 5.2.** *If $k \geq 2$ and a concave function $f : P_{n,k} \to [0, 1]$ is accessible through a relative-$\delta$-erroneous oracle, then for any $\epsilon \in [0, \delta]$, an algorithm that makes less than $e^{\epsilon^2 n/6k}$ queries cannot find a solution of value greater than $\frac{1+\epsilon}{k}OPT$ with probability more than $2e^{-\epsilon^2 n/6k}$.*

Note that this result is nontrivial only for $n \gg k$. In other words, the hardness factor of $k$ is never worse than a square root of the dimension of the problem. Therefore, this result can be viewed as interpolating between the hardness of $\frac{1+\epsilon}{2}$-approximation over the unit cube (Theorem 3.2), and the hardness of $n^{\delta-1/2}$-approximation over a general polytope (Theorem 4.2).

*Proof.* Given $\pi : [n] \to [k]$, we construct a function $f^\pi : P_{n,k} \to [0, 1]$ (where $\pi$ describes the intended optimal solution):

- $f^\pi(\mathbf{x}) = \frac{1}{n}\sum_{i=1}^{n} x_{i,\pi(i)}$.

Next we define a modified function $\tilde{f}^\pi$ as follows:

- If $|f^\pi(\mathbf{x}) - \frac{1}{k}| > \frac{\epsilon}{k}$ then $\tilde{f}^\pi(\mathbf{x}) = f^\pi(\mathbf{x})$

- If $|f^\pi(\mathbf{x}) - \frac{1}{k}| \leq \frac{\epsilon}{k}$ then $\tilde{f}^\pi(\mathbf{x}) = \frac{1}{k}$.

By definition, $f^\pi(\mathbf{x})$ and $\tilde{f}^\pi(\mathbf{x})$ differ only if $|f^\pi(\mathbf{x}) - \frac{1}{k}| \leq \frac{\epsilon}{k}$, and then $f^\pi(\mathbf{x}) \in [\frac{1-\epsilon}{k}, \frac{1+\epsilon}{k}]$ while $\tilde{f}^\pi(\mathbf{x}) = \frac{1}{k}$. Therefore, $\tilde{f}^\pi(\mathbf{x})$ is a valid relative $\epsilon$-erroneous oracle for $f^\pi$.

Similarly to the proofs above, we argue that if $\pi$ is chosen uniformly at random, then with high probability $\tilde{f}^\pi(\mathbf{x}) = \frac{1}{k}$ for any fixed query $x \in P_{n,k}$. This holds again by a Chernoff bound: For a fixed $x_{ij}$ such that $\sum_{j=1}^{k} x_{ij} = 1$, we have that $f^\pi(\mathbf{x}) = \frac{1}{n}\sum_{i=1}^{n} x_{i,\pi(i)} = \frac{1}{n}Z$ where $Z$ is a sum of independent random variables with expectation $\frac{1}{k}\sum_{i,j} x_{ij} = \frac{n}{k}$. The random variables attain values in $[0, 1]$. By the Chernoff bound, $\Pr[|Z - \frac{n}{k}| > \epsilon\frac{n}{k}] < 2e^{-\epsilon^2 n/3k}$. This gives

$$\Pr\left[|f^\pi(\mathbf{x}) - \frac{1}{k}| > \frac{\epsilon}{k}\right] < 2e^{-\epsilon^2 n/3k}.$$

By the same arguments as before, if the algorithm asks less than $e^{\epsilon^2 n/6k}$ queries, then it will not detect a point such that $|f^\pi(\mathbf{x}) - \frac{1}{k}| > \frac{\epsilon}{k}$ with probability more than $2e^{-\epsilon^2 n/6k}$. Then the query answers will all be $\tilde{f}^\pi(\mathbf{x}) = \frac{1}{k}$ and the value of the returned solution will be at most $\frac{1+\epsilon}{k}$. Meanwhile, the optimum solution is $x_{i,\pi(i)}^* = 1$ for all $i$, which gives $f(\mathbf{x}^*) = 1$. $\square$

**Acknowledgements.** YS was supported by NSF grant CCF-1301976, CAREER CCF-1452961 and a Google Faculty Research Award.

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
