[Reviews · NeurIPS 2015]

Submitted by Assigned_Reviewer_1

This being a light review (and myself being an optimisation person, rather than an information theorist), I have not checked the proofs in detail.

Most computations on modern machine suffer from the double precision (or worse). Most analyses of iteration complexity gloss over that, whereas this paper exposes the peril of error in certain oracles.

The profs are based on refs. [6,15] of Vondrak and team, which deal with unconstrained submodular maximization. The realisation that this carries over is novel and very important, though.
Summary: This is an important, strikingly original contribution, provided the proofs are correct.

Submitted by Assigned_Reviewer_2

Theorem 3.1 is very nice. At first I was skeptical, and then after reading the proof it became obvious why this was true. A natural question is whether one can bridge the gap between this negative result and the positive result of Belloni et al. It would be nice if the authors made a conjecture. I personally believe that Belloni et al. result should be true at epsilon / sqrt (n) (instead of epsilon / n currently), which would essentially match the lower bound of Theorem 3.1.
Summary: Surprising answer to an interesting question.

Submitted by Assigned_Reviewer_3

Added post-rebuttal: Thanks for addressing my technical concerns, and I've raised my score to 6. Note that the \delta,\epsilon issue still seems a bit weird (for example, theorem 4.2, why not just take delta=0.5, getting the strongest possible statement?).

============================================

The paper studies lower bounds for zero-order optimization, where the returned function values have some arbitrary bounded error. Previous results provided algorithms when the error scales down linearly in the dimension n. In contrast, here it is shown that if the error scales more slowly than n^{-1/2}, then no non-trivial optimization is possible. This is shown in several settings, considering both absolute and relative errors, and for several different optimization domains. The proof technique is based on constructing two functions which agree almost everywhere on the domain, yet have quite different optima.

I am a bit undecided regarding this paper. On one hand, the basic question is scientifically interesting, and the answer is somewhat surprising at first, and potentially useful. On the other hand, there are several weaknesses:

- Beyond the first interesting result (theorem 3.1), the other results are variations of the same idea for various specific domains (whose relevance for zero-order optimization is unclear), and I didn't feel they provide anything conceptually new. I think the paper would have been stronger if it contained more genuinely different results. For example, is it possible to generalize the impossibility results here to general families of domains? What is the error in the oracle values is between n^{-1/2} and n^{-1} (at which point existing algorithms do work)? What if the oracle error isn't arbitrary but has some structure?

- The connection of the paper to machine learning is somewhat tenuous (in previous papers, which considered a noisy oracle, one can at least argue that such an oracle reflects uncertainty due to data sampling).

- Although I believe the results, the proofs are sloppy at places and some parts need to be fixed (see technical comments below).

- The text is wider than the required format (maybe because of an erroneous use of \fullpage in the latex?) This means that the paper will require some non-trivial editing to get to the page limit.

Specific Comments ----------------- - All the proofs rely on extending a Chernoff bound for a fixed query to a sequence of several queries, using a union bound. However, after the first query, the other queries are no longer fixed, but rather conditioned on previous queries. So, one cannot apply a Chernoff bound as-is, and a more delicate argument is required.

- In theorem 3.1, one should also require delta < 1, no? Otherwise the proof doesn't seem to work. Similarly, in theorem 4.2, I think delta must be < 0.5 (otherwise, in the proof, f and g would be the same function).

- In all the proofs, there is an implicit assumption that n^{delta} and such numbers are integers (e.g. splitting [n] to sets of this size), which is not necessarily the case.

- In the proof of theorem 4.2, the definition of the polytope domain is missing (and is apparently not the same as in theorem 4.1, since the optimal values of f and g would be different than those stated in the proof).

- In the proof of theorem 3.1, the fact that the function value is the sum of negatively correlated random variables needs to be more carefully justified, either by a reference or by explicitly showing why they satisfy the definition of negative correlation from section 2.

- The results of section 4 appear to be stronger the larger we pick epsilon,delta. So, why not show the result for some large constant epsilon,delta.?

- Section 5 is extremely short and appears to be just an advertisement to something which appears in supplementary material. This makes the paper not self-contained. I would recommend either providing some formal result (possibly without the proof), or dropping this section.

- line 319: "We again we select"
Summary: The paper provides an interesting and potentially useful observation about the effect of error in zero-order convex optimization. However, it's a bit thin on new ideas, and has some issues with the proofs and format (although these are potentially fixable).

Submitted by Assigned_Reviewer_4

You get lots of mileage from the nice construction, but the paper might be even better by exposing it front and center, rather than hiding it in a proof of highly abstract theorems.

I think you could probably enhance the relevance by considering the precision to which the optimum can be estimated (in reasonable queries) for any given dimension and error level, rather than the more synthetic setting you consider.

Minor comments: - The dashed line in Figure 1 (the oracle answers, presumably?) enjoys multiple values for some x, not clear what this says about the powers of the oracle. Also the vertical error margins, as drawn vary wildly with x.
Summary: Paper turns a well known fact about concentration of sums into a still somewhat surprising fact about nearly flat functions as seen through disturbed function values. Reasonably relevant to NIPS, and very nicely written.

Author Feedback
Author rebuttal: We thank all reviewers for their positive feedback and suggestions.

Reviewer nr 1:
Thank you for the positive feedback. At this point we are still a bit reluctant to make a conjecture, but hopefully we will develop better intuition soon, and if relevant add a conjecture to the manuscript.

Reviewer nr 2:
Thank you for the detailed review.

- Regarding more results:

a. other general families of domains: please note that we prove the result also holds for strongly convex functions in the last section of the paper; also note that the functions for which the lower bound holds are quite simple (affine functions) that are included in many other function classes.

b. Closing the gap between n^{-1} and n^{-1/2}: agreed; see response to reviewer #1;

c. Oracle having some structure: if the erroneous oracle happens to be a convex function, positive results are again obtainable; we can add this remark in the manuscript.

Connection to machine learning: our view is that errors may in some cases be inevitable; the i.i.d random noise model is a certainly potent modeling tool, though there are some cases that it may be inappropriate. In cases when data is corrupted (as pointed out by reviewer 5 for example), erroneously collected, or if the objective is approximated from data, the i.i.d assumption may be too strong. We can discuss this in further detail in the manuscript.

Sloppy proofs: we don't believe this is the case, please see our remarks.

- adaptive queries: we address the issue of adaptive queries and randomized algorithms. please see lines 231 -- 250 inside proof 3.1 in the manuscript;

- n is not integer: this is for notational brevity, one can round to nearest integer;
we will add a remark in the preliminaries.

- \delta <1 in 3.1: yes, \delta is always <1; we will add this.

- \delta < 0.5 in Thm 4.2: this is not necessary; \delta does not need to be strictly smaller than 0.5 since in f the sum is only on i \in H while in g the sum is on all i\in [n].

- definition of polytopes in Thm 2: The definition is not missing. Please see line 330 where we specify b (the polytope constraint as defined in line 262), and the functions' values under this constraint.

- negatively correlated random variables: we will explain in greater detail how claim 2.1 is applied to sampling sets u.a.r rather than elements i.i.d;

- wider text margins: we will fix this issue, and if necessary defer some proof details to a full version that we will make available on the arXiv, which will include all supplementary material.

- section 5: thank you for the suggestion, we will try to provide a formal statement in this section.

-typo in line 319: thanks, we will fix this typo.

Reviewer nr 3:
- We will state explicit bounds as a dependency of the dimension;
- We will fix the illustration so that f(x) only takes one value and state that the grey area indicates the range of error \epsilon

Reviewer nr 4:
- Thank you for the positive feedback!

Reviewer nr 5:
- Thank you for the positive feedback, and thank you for pointing out the application to error due to double precision, we will add this point to the discussion.

Reviewer nr 6:
Thank you for the positive feedback. The question of what lies between the oracle models you mention and those in this paper is certainly interesting, and we intend to pursue it in future work.

To all reviewers:
- We will add remarks and short discussion to address comments above.
- We will also add two references to the related work section that are currently not in the manuscript. The first is to a relevant example by Nemirovsky and Yudin ("Problem Complexity and Method Efficiency in Optimization", Chapter 9) which shows a related model of error in zeroth order oracle and informally argues that exponentially many queries are required in that case. The second reference is to information theoretic lower bound in submodular maximization with erroneous oracles in an unpublished manuscript of recent work by Avinatan Hassidim and Yaron Singer. In this case the oracle is for submodular functions and the error is multiplicative. Hassidim and Singer use the example to motivate the study of submodular maximization under matroid constraint with random (as opposed to adversarial) noise. In both cases, the models and motivation differ from this work. We do not think that these works affect the contribution of the paper, but we feel we should add these references to the related work section.